# Impact of disease stage and age at Parkinson's onset on patients' primary concerns: Insights for targeted management

**Roongroj Bhidayasiri**[1,2]*, **Thanatat Boonmongkol**[1], **Yuwadee Thongchuam**[1], **Saisamorn Phumphid**[1], **Nitinan Kantachadvanich**[1], **Pattamon Panyakaew**[1], **Priya Jagota**[1], **Rachaneewan Plengsri**[3], **Marisa Chokpatcharavate**[3], **Onanong Phokaewvarangkul**[1]

1 Chulalongkorn Centre of Excellence for Parkinson's Disease and Related Disorders, Department of Medicine, Faculty of Medicine, Chulalongkorn University and King Chulalongkorn Memorial Hospital, Thai Red Cross Society, Bangkok, Thailand, 2 The Academy of Science, The Royal Society of Thailand, Bangkok, Thailand, 3 Chulalongkorn Parkinson Patients' Support Group, Chulalongkorn Centre of Excellence for Parkinson's Disease and Related Disorders, Bangkok, Thailand

* rbh@chulapd.org

## Abstract

### Background

The concerns of people with Parkinson's disease (PD) about their disease are often different from the objective clinical picture and subject to various influencing factors, including disease progression. Currently our understanding of these concerns is limited, particularly in Asian countries.

### Methods

A 50-item survey on Parkinson's Disease Patients' Concerns (PDPC Survey) was developed by a multidisciplinary care team. The subjective greatest concerns (most commonly concerning symptoms) of patients at a specialist centre in Bangkok, Thailand, were explored and categorised according to disease stage and age at onset of PD.

### Results

Data for 222 patients showed concerns varied widely. Motor symptoms giving the greatest concern were problems with walking and/or balance (40.5% of patients), while the most commonly concerning non-motor symptom (NMS) was constipation (41.0%). Patterns were observed amongst different patient subgroups. Early PD patients (H&Y stage 1) were more concerned about NMS than motor symptoms, while the reverse was true for advanced PD patients. Young-onset PD patients showed significantly greater concerns than typical-onset patients about motor symptoms relating to social functioning, working and stigmatisation, such as speech ($p = 0.003$).

### Conclusions

This study, in an Asian patient cohort, provides an assessment of a wide range of PD patients' concerns, encompassing not only motor symptoms and NMS, but also treatment-

**Data Availability Statement:** All relevant data are within the manuscript and its Supporting Information files.

**Funding:** RB is supported by Senior Research Scholar Grant (RTA6280016) of the Thailand Science Research and Innovation (TSRI), International Research Network Grant of the Thailand Research Fund (IRN59W0005), and Center of Excellence grant of Chulalongkorn University (GCE 6100930004-1).

**Competing interests:** There is no conflict of interest on all authors and we will take full responsibility for the data, the analyses and interpretation, and the conduct of the research.

related adverse events, care in the advanced stage, and the need for assistive devices. Identifying the concerns of individual PD patients and implementing a patient-centred approach to care is critical to their wellbeing and optimal outcomes. The PDPC survey can help healthcare teams build a more accurate picture of patients' experiences to inform clinical management.

## Introduction

The underlying concerns of people with Parkinson's (PD) about their condition can often be different from the objective clinical symptoms they present to the healthcare team. In a broad sense, concerns reflect what we believe or perceive and, in the setting of PD, could be current problems, future worries, both, or even something in between. Moreover, patients' concerns may change or fluctuate, depending on a range of influencing factors, including age at onset of PD, disease severity, disease progression, prevailing social or cultural circumstances, and their personal subjective experiences [1–6]. Nowadays, it is common for patients to seek information from websites and social media, which can also influence their perceptions and understanding of PD and its treatment. How specific factors contribute to each patient's view is likely to be complex and will vary according to individual circumstances. In addition, knowledge gaps or the level and accuracy of patients' understanding of their disease may influence their attitudes and opinions [7]. Patients' concerns about possible stigmatisation can also alter their view of their condition and its treatment, and even lead to social isolation [8, 9].

Various centres have undertaken surveys of PD patients perceptions and what might influence them but, overall, data are limited. Studies report a wide diversity of patient experiences and note the significant impact of non-motor symptoms (NMS), in addition to motor issues [1, 2]. In some cases a disconnect between physicians' and patients' perceptions of their disease and its treatment has been noted [10], highlighting the need for a deeper understanding of patients' experiences. The group 'Parkinson Inside Out' includes healthcare professionals and researchers who have been diagnosed with PD [11, 12] and are therefore in the unique position of understanding what the disease looks like from a clinical standpoint as well as what it feels like to live with PD every day. They highlighted pain, sleep disturbances, and impulse-control disorders as key issues. Cultural differences in PD patients' perceptions of treatment have been reported, for example between the USA and Japan [6] however specific studies in Asian countries are lacking. A small study from Taiwan found common themes in the patients' experiences, namely a lack of knowledge about of symptoms, feeling a loss of control, gradual deterioration, and a deep sense of helplessness [13]. A larger survey from Thailand found significant knowledge gaps amongst PD patents across three domains: diagnosis, therapeutic options, and disease course, suggesting the need for education [7].

The challenge for healthcare providers is that the types of concerns expressed by PD patients are often subjective aspects of the disease experience that are not captured with traditional diagnostic tools used in the clinic. While standard rating scales, such as the Movement Disorders Society's Unified Parkinson's Disease Rating Scale [14], are valuable clinical tools, they can lack the granularity to detect the individual variations in disease manifestation and the more subjective aspects of a patient's disease experience.

Patient-reported outcomes (PROs) relating to symptoms and health-related quality of life (HRQoL) are a direct reflection of the patient's perceived health status and their overall wellbeing. PROs can therefore provide important additional information to help build a more

accurate clinical picture. It is important that physicians are aware of the patient's perspective of their symptoms and understand their impact on that person so they can fully engage, and communicate well with them, their family and caregivers, as well as being able to target the right strategy to address their concerns [5].

Efforts have been made to develop patient-centred measures of symptom severity in PD using self-rating scales to assess the patient's condition, which have shown good correlation with some existing clinical scales [15]. However, most investigations have focused on motor and non-motor aspects of the disease and lack information about other dimensions that may also influence HRQoL. Therefore, our study aimed to explore patient's greatest concerns about a range of factors, such as their overall feeling of happiness, the impact of adverse events, and issues regarding palliative care and assistive devices, in addition to motor symptoms and NMS. Variations in these concerns amongst different groups of patients–according to their stage of disease and age of onset of PD–were also evaluated with the aim of providing information to enable healthcare providers to target the most suitable PD care and management strategy to each group of patients.

## Methods

### Survey development and steps of validation

A multidisciplinary care team at Chulalongkorn Centre of Excellence for Parkinson's Disease and Related Disorders (ChulaPD, www.chulapd.org), comprising movement disorder neurologists, PD Nurse Specialists, physical therapists, social workers, occupational therapists, representatives from a PD support group (both patients and caregivers), worked in collaboration to determine the process of constructing a new survey related to patients' concerns in English language. All members were bilingual and all healthcare professionals had extensive experience, of at least 5 years, in the care of PD patients. All members were requested to review key publications and scales to determine PD-specific themes that are related to the following domains: 1) motor symptoms; 2) non-motor symptoms (NMS); 3) symptoms of fluctuations; 4) adverse events; 5) care in the advanced stage; and 6) assistive devices as well as happiness level [14, 16–28]. In addition, one-on-one and group interviews were also conducted with patients and caregivers attending a local PD support group. These interviews were structured and unscripted. The information from these sources guided the development of PD-specific items, which finally generate 50 items for the survey on Parkinson's Disease Patients' Concerns (PDPC Survey) covering these six domains and happiness level (S1 Table). Each item was assigned an ordinal response based on a severity scale from 0 (not concerned) to 10 (most concerned). It includes questions on subjective happiness (1 item), motor symptoms (12 items), NMS (13 items), symptoms of fluctuations (wearing off and dyskinesia; 9 items), adverse events (AEs) related to treatment (6 items), care in the advanced stage (7 items), and the use of assistive devices (2 items), and takes approximately 15–30 minutes to complete.

Comprehension of all items were tested for content validity by another expert panel who were not involved with item generation. The index of item-objective congruence (IOC) was conducted on all questionnaire items, demonstrating a positive content validation of all IOC index of at least 0.6 on all items (S1 Data). The PDPC Survey was then translated, according to the translation standards, into Thai (PDPC Survey-Thai version) and back-translated into English, with modifications of the Thai wordings where necessary. The consistencies of both Thai and English versions were then approved by two independent bilingual movement disorder neurologists who were not involved in the PDPC Survey development. Other aspects of survey development and validation were not performed.

## Administration of the survey on Parkinson's disease patients' concerns

Between March and June 2019, the PDPC Survey-Thai version was administered by one of the healthcare team to PD patients attending outpatient clinics of the ChulaPD, which is a main tertiary referral centre for PD, affiliated with Chulalongkorn University Hospital and the Thai Red Cross society. Patients with cognitive impairment (Mini Mental State Examination score <23) were excluded. Patients were asked to rate their concerns for each item on a severity scale from 0 (not concerned) to 10 (most concerned) based on their experience during the past month. The severity was further classified as 'Some concern' for the rating between 0 and 6 and 'Most concerning' for the rating between 7 and 10. The distinction between 'Some concern' and 'Most concerning' was determined by the consensus of the Chulalongkorn Parkinson Patients' Support Group, represented by two authors (RP and MC), who perceived that the severity for 'Most concerning' should reflect the 30% end of the severity spectrum. Results were reported in a form of most commonly concerning symptoms, referring to what the most patients rated these symptoms as highly concerning. Disease stage was assessed using the Hoehn and Yahr (HY) scale [29]. The presence of caregiver in this study refers to a stable main caregiver, which is defined as any person who, without being a professional or belonging to a social support network, usually lives with the patient and, in some way, is directly implicated in the patient's care or is directly affected by the patient's health problem [30]. Since there is no consensus on the definition of early and advanced stages of PD, we adopted the traditional classification of early and advanced stages according to the HY scale where bilateral segmental involvement with postural and gait impairment (HY stage > 3), which also marks a clinical milestone, classifies advanced stage whereas early stage refers to those with HY stage 1–2 [31]. The study was approved by the Human Ethics Committee of the Faculty of Medicine, Chulalongkorn University (IRB. No. 134/62). All participants gave informed written consent before entering the study in accordance with the Declaration of Helsinki. Data were collected in Excel files, encrypted, anonymised, and stored on ChulaPD's secure data server for analysis.

## Statistical analysis

Patient demographics and baseline characteristics were summarised using either means and standard deviations (SD) or frequencies and percentages, as appropriate. Chi-squared tests were used to compare the differences for categorical variables between three subgroups: (1) PD patients with or without caregivers, (2) early (HY 1–2) or advanced stage of PD (HY ≥3), and (3) young-onset (onset of symptoms at <50 years of age) or typical-onset PD. An unpaired t-test was used to compare the differences for continuous variables where the data were fit to a normal distribution for the comparison of the above three subgroups where data was fitted to a normal distribution. In addition, an effect size for mean differences of the PDPC survey scores of PD patients with and without caregivers was calculated for Cohen's d.

To identify predictors of having a caregiver, binary logistic regression analysis was performed in which the presence of caregiver need was a dependent variable and twenty participant-related variables were selected to run into the logistic model as independent variables, including problems with walking and/or balance, getting out of bed, freezing of gait, constipation, daytime sleepiness, pain and/or aching, slow movement, muscle cramping, difficulty performing fine finger movements, any stiffness, marked decline in physical ability, cognitive difficulties (part of NMS), bedridden wheelchair bound, difficulty swallowing, cognitive difficulties (part of symptoms of the advance stage), age, gender, presence of postural instability, and presenting symptoms of bradykinesia. Goodness-of-fit statistics using the Hosmer–Lemeshow test helped to determine whether the model adequately described the data and indicated a poor fit if the significance value was <0.05. The logistic model was undertaken with Forward (Wald) stepwise technique in order to

select the most predicable variables to determine caregiver need in PD patients. The predictors were reported as odds ratio (OR) with a $p$-value of $<0.05$ (2-tailed) considered statistically significant. All statistical analyses were performed using SPSS version 23.0 software.

## Results

The PDPC Survey was completed by 222 PD patients. Patients included in the analysis had a mean age of 68 years and a mean H&Y scale score of 2.76; half of the patients (50.5%) had stable caregivers (Table 1).

### Total patient cohort

Overall, the subjective happiness rating on a scale from 0 (very unhappy) to 10 (very happy) was a mean (SD) of 6.41 (2.23). Responses regarding motor symptoms showed those rated as giving the most commonly concern were problems with walking and/or balance (40.5%), getting out of bed (39.6%), and freezing of gait (36.5%) (Table 2). For NMS, constipation represented the symptom that gave the most commonly concern (41.0%) (Table 2). For symptom fluctuations, slow movement (41.4%; an OFF-period symptom) was the symptom that gave the most commonly concern, followed by muscle cramping (36.5%) and any stiffness (34.7%) (Table 2). Drug-induced dyskinesia was not rated by patients as a symptom of most commonly concern.

In terms of possible treatment-related AEs, symptoms that were rated as the most commonly concerning were GI (31.5%), urinary (23.9%) and neuropsychiatric (16.7%) (Table 2).

### Concerns in advanced stage

For symptoms related to the care in the advanced stage, the three symptoms that rated as the most commonly concerning were a marked decline in physical ability (40.5%),

**Table 1. Demographics of the 222 PD patients who completed the Parkinson's disease patients' concerns survey.**

| Item | Results |
|---|---|
| Age | 67.69 ± 10.84 |
| Male gender | 125 (56.3%) |
| Disease duration | 10.40 ± 6.58 |
| Presenting symptoms at the time of diagnosis: | |
| ▪ Bradykinesia | 1. (61.7%) |
| ▪ Tremor | 1. (48.6%) |
| ▪ Rigidity | 1. (41.9%) |
| ▪ Gait difficulty | 87 (39.2%) |
| Mean HY Score (all patients) | 2.76 ± 0.91 |
| Number at each HY stage: | |
| ▪ Stage 1 | 1. (2.7%) |
| ▪ Stage 2 | 1. (43.7%) |
| ▪ Stage 3 | 1. (32.9%) |
| ▪ Stage 4 | 1. (16.2%) |
| ▪ Stage 5 | 10 (4.5%) |
| Presence of postural instability | 119 (53.6%) |
| Presence of caregivers | 112 (50.5%) |

Categorical data are reported as number and percentage; continuous data are reported as mean ± standard deviation. HY: Hoehn & Yahr stage.

**Table 2.** PD patients' greatest concerns regarding (a) motor symptoms, (b) non-motor symptoms, (c) symptoms fluctuations, (d) adverse events, and (e) palliative care (n = 222).

| Most commonly concerning symptoms | Percentage of patients reporting most commonly concerning symptoms | Overall severity |
|---|---|---|
| **a. Motor symptoms** | | |
| Problems with walking and/or balance | 40.5% | 5.45 ± 2.93 |
| Getting out of bed | 39.6% | 5.10 ± 3.07 |
| Freezing of gait | 36.5% | 4.92 ± 3.39 |
| **b. Non-motor symptoms** | | |
| Constipation | 41.0% | 5.16 ± 3.39 |
| Fatigue | 31.1% | 4.40 ± 2.98 |
| Urinary problems | 27.5% | 4.15 ± 3.15 |
| **c. Symptom fluctuations** | | |
| Slow movement | 41.4% | 5.21 ± 3.06 |
| Muscle cramping | 36.5% | 4.80 ± 3.11 |
| Any stiffness | 34.7% | 4.61 ± 3.17 |
| **d. Adverse events** | | |
| Gastrointestinal symptoms | 31.5% | 4.51 ± 3.22 |
| Urinary symptoms | 23.9% | 3.96 ± 3.27 |
| Neuropsychiatric symptoms | 16.7% | 3.05 ± 2.96 |
| **e. Care in the advanced stage** | | |
| Marked decline in physical ability | 40.5% | 5.42 ± 3.22 |
| Bedridden/wheelchair bound | 35.1% | 4.47 ± 3.88 |
| Cognitive difficulties | 30.2% | 4.22 ± 3.29 |

Severity scores are reported as mean ± standard deviation. Symptoms were classified as 'most concerning' for patient's ratings of 7 and beyond.

becoming bed- or wheelchair-bound (35.1%), and cognitive difficulties (30.2%). The predominant concerns in the advanced stage were factors found to be associated with having a caregiver, namely chewing and swallowing, and getting out of bed, both of which had statistically significant odds ratios (Tables 3 and S2). The presence of caregivers was significantly higher amongst those with advanced PD compared with early PD patients ($p<0.01$). In a comparison of PDPC Survey scores between patients with caregivers and those without, a similar pattern was observed with factors such as difficulty chewing and swallowing, getting out of bed, and problems with walking and/or balance, rated as having a significantly higher severity by those with caregivers (S2 Table).

**Table 3. Factors associated with presence of a caregiver.**

| Predictive factor | Model: Odds ratios/Exp(B) |
|---|---|
| **Difficulty chewing and swallowing** | 2.455* |
| **Getting out of bed** | 3.751* |
| ***Hosmer–Lemeshow test*** | *0.936* |

*Statistically significant using the binary logistic model.

## Analysis by Hoehn & Yahr stage

When patients' greatest concerns about motor symptoms and NMS were analysed according to their HY stage, it was found that patients at HY stage 1 generally did not report great concern about motor symptoms whilst patients at more advanced HY stages described symptoms of greatest concern relating to problems with walking and/or balance, freezing of gait, and getting out of bed (Table 4). On the other hand, patients at HY stage 1 reported greatest concern about NMS such as constipation, lack of interest or enthusiasm, or urinary problems. Overall, constipation was rated as the most commonly concerning NMS. While patients at HY stage 1 did not report great concerns about motor symptoms, they shared some concerns on problems with walking and/or balance in 66% of patients, followed by difficulty speaking and shaking both in half of patients.

## Analysis by age of onset of PD

When comparing the greatest concerns of young-onset PD patients with those of typical-onset PD patients, our study found that young-onset patients rated freezing of gait as the most commonly concerning motor symptom (46.3%), whereas for typical-onset PD it was getting out of bed, and problems with walking and/or balance (both 39.5% of patients) with freezing of gait ranked second (33.5%) (Table 5). Constipation was rated as the most commonly concerning NMS by both patient cohorts. In terms of symptom fluctuations, the highest ranking most commonly concerning symptoms in young-onset PD patients were slow movement (57.4%), difficulty performing fine finger movements (53.7%), any stiffness (50.0%), and muscle cramping (50.0%), possibly because these are factors that relate to social functioning, working and stigmatisation. A similar profile of symptoms was observed in the typical-onset PD group, however there was a trend to lower numbers of patients reporting them, and with numerically lower severity ratings.

Even though drug-induced dyskinesia was not rated amongst the top three most commonly concerning symptoms by either group of patients, young-onset PD patients reported drug-induced dyskinesia as a most commonly concerning symptom significantly more than typical-onset PD patients (33.3% vs 9%, $p < 0.001$).

**Table 4. Top three symptoms of most commonly concern on (a) motor symptoms and (b) non-motor symptoms according to Hoehn and Yahr stage (n = 222).**

| Stage 1 | Stage 2 | Stage 3 | Stage 4 | Stage 5 |
|---|---|---|---|---|
| **a. Motor symptoms** | | | | |
| None | Getting out of bed (35.1%) | Problems with walking and/or balance (46.6%) | Problems with walking and/or balance (58.3%) | Freezing of gait (80.0%); Getting out of bed (80.0%); Turning in bed (80.0%) |
| None | Problems with walking and/or balance (28.9%) | Freezing of gait (37.0%) | Freezing of gait (55.6%); Getting out of bed (55.6%) | Problems with walking and/or balance (70.0%) |
| None | Freezing of gait (26.8%) | Getting out of bed (35.6%) | Turning in bed (44.4%) | Difficulty speaking (60.0%); Saliva and drooling (60.0%); Social activities (60.0%) |
| **b. Non-motor symptoms** | | | | |
| Constipation (16.7%) | Constipation (36.1%) | Constipation (49.3%) | Constipation (41.7%); Insomnia (41.7%) | Urinary problems (70.0%) |
| Lack of interest or enthusiasm (16.7%) | Daytime sleepiness (29.9%) | Urinary problems (31.5%) | Fatigue (36.1%) | Fatigue (60.0%) |
| Urinary problems (16.7%) | Fatigue (28.9%) | Fatigue (30.1%) | Pain and other sensation (33.3%) | Constipation (40.0%); Daytime sleepiness (40.0%); Hallucination and delusions (40.0%); Lack of self-control (40.0%) |

Data were categorised according to number and percentage. No additional items were included in the 2nd and 3rd ranking for NMS under HY stage 1 patients due to equal scoring of top three symptoms in the 1st ranking.

**Table 5. Comparison of the most commonly concerning symptoms between young-onset and typical-onset PD patients on (a) motor symptoms, (b) non-motor symptoms, (c) symptom fluctuations, (d) adverse events, and (e) care in the advanced stage (n = 222).**

| Young-onset PD | | Typical-onset PD | |
| --- | --- | --- | --- |
| Symptom (%) | Overall severity | Symptom (%) | Overall severity |
| **a. Motor symptoms** | | | |
| Freezing of gait (46.3%) | 5.37 ± 3.40 | Getting out of bed (39.5%) | 5.04 ± 3.10 |
| | | Problems with walking and/or balance (39.5%) | 5.31 ± 2.94 |
| Problems with walking and/or balance (44.4%) | 5.87 ± 2.91 | Freezing of gait (33.5%) | 4.77 ± 3.39 |
| Difficulty speaking (42.6%) | 5.65 ± 2.99 | Turning in bed (29.3%) | 3.33 |
| **b. Non-motor symptoms** | | | |
| Constipation (48.1%) | 5.67 ± 3.19 | Constipation (38.9%) | 4.99 ± 3.46 |
| Fatigue (31.5%) | 4.81 ± 3.03 | Fatigue (31.1%) | 4.26 ± 2.97 |
| Daytime sleepiness (27.8%) | 4.22 ± 2.75 | Urinary problems (29.9%) | 4.25 ± 3.22 |
| Anxiety and/or panic attacks (27.8%) | 4.81 ± 2.50 | | |
| **c. Symptom fluctuations** | | | |
| Slow movement (57.4%) | 6.46 ± 2.62 | Slow movement (36.5%) | 4.80 ± 3.10 |
| Difficulty performing fine finger movements (53.7%) | 6.26 ± 2.72 | Muscle cramping (31.7%) | 4.41 ± 3.13 |
| | | Pain and/or aching (31.7%) | 4.66 ± 2.89 |
| Any stiffness (50.0%) | 5.76 ± 3.05 | Any stiffness (29.3%) | 4.23 ± 3.13 |
| Muscle cramping (50.0%) | 5.96 ± 2.75 | | |
| **d. Adverse events** | | | |
| Gastrointestinal symptoms (37%) | 4.72 ± 3.40 | Gastrointestinal symptoms (29.9%) | 4.47 ± 3.17 |
| Urinary symptoms (31.5%) | 4.35 ± 3.41 | Urinary symptoms (21.6%) | 3.84 ± 3.23 |
| General symptoms (16.7%) | 3.44 ± 2.85 | Neuropsychiatric symptoms (16.8%) | 2.95 ± 2.98 |
| Neuropsychiatric symptoms (16.7%) | 3.33 ± 2.91 | | |
| **e. Care in the advanced stage** | | | |
| Marked decline in physical ability (53.7%) | 6.63 ± 2.84 | Marked decline in physical ability (35.9%) | 5.01 ± 3.24 |
| Bedridden/wheelchair bound (44.4%) | 5.26 ± 4.05 | Bedridden/wheelchair bound (31.7%) | 4.18 ± 3.78 |
| Difficulty swallowing (42.6%) | 4.67 ± 3.61 | Cognitive difficulties (28.1%) | 3.95 ± 3.29 |

Severity scores are reported as mean ± standard deviation.

The predominant treatment-related AEs of greatest concern were GI and urinary symptoms in both patient groups. The top two ranked most commonly concerning symptoms regarding care in the advanced stage in both groups were a marked decline in physical ability and being bedridden/wheelchair bound, however higher numbers on the young-onset PD group reported these and there was a trend to higher rankings of severity than in the typical-onset PD group.

S3 Table shows a comparison of the severity of different concerns of young-onset versus typical PD patients regarding motor symptoms, NMS, and symptom fluctuations. Patients with young-onset PD reported significantly more concerns about a number of motor symptoms and symptom fluctuations, including difficulty speaking ($p = 0.003$), eating tasks ($p = 0.003$), washing and bathing ($p = 0.04$), shaking ($p = 0.005$), and dyskinesia ($p = 0.001$) and a number of NMS, including low and/or depressed mood ($p = 0.01$) and anxiety and/or panic attacks ($p<0.001$) when compared to typical-onset PD patients.

## Need for assistive devices

Analysis of responses for the total patient cohort regarding concerns about the need for assistive devices found that a shower chair (38.3%) was considered as the most needed device

amongst, followed by a walking stick (36.5%) and anti-slip mat (34.7%) (Table 6). When results were analysed according to whether patients had early-stage or advanced-stage PD, advanced-stage patients reported a significantly greater need for a walking stick ($p = 0.01$), a wheelchair ($p<0.001$), and an anti-choking cup ($p = 0.03$) than early-stage patients (Table 6).

## Discussion

Our study using the PDPC survey found that patients' concerns about their symptoms vary widely and depend on how they perceive their motor symptoms, NMS, fluctuations, and over-all treatment experience, at different disease stages, which also reflects in the results of previous studies [1, 2]. However, although symptoms in PD are individualised, we also observed a pattern of concerns amongst different subgroup of PD patients (young-onset PD versus typical onset PD and early versus advanced PD).

Some patients' greatest concerns reflect their current symptoms whilst others relate more to worries of what will happen to them in the future, as shown when results were analysed according to patients' HY stage. In our study, those at HY stage 2 were concerned about getting out of bed, gait/balance, and freezing, which are generally not the symptoms that manifest at this stage, and suggests that they may be worrying about developing these disabilities in the future. A previous study undertaken to assess the level of PD severity associated with disability and to identify the sequence of loss of independence in common daily activities found that disability with loss of independent function was found at HY stages 2–3 [32]. Difficulty with daily activities, without loss of independent function was reported earlier, at HY stages 1–2, so

**Table 6. Patients' concerns regarding the need for assistive devices: Comparison of patients with early-stage versus advanced-stage PD.**

| Item | All participants (n = 222) | Early-stage PD (n = 103) | Advanced-stage PD (n = 119) | p-value |
|---|---|---|---|---|
| Walking stick | 81 (36.5%) | 29 (28.2%) | 52 (43.7%) | 0.016* |
| Laser-guided walking stick providing visual cues | 42 (18.9%) | 19 (18.4%) | 23 (19.3%) | 0.867 |
| Walker | 40 (18.0%) | 14 (13.6%) | 26 (21.8%) | 0.110 |
| Wheelchair | 57 (25.7%) | 14 (13.6%) | 43 (36.1%) | <0.001* |
| Electric adjustable bed | 44 (19.8%) | 19 (18.4%) | 25 (21.0%) | 0.633 |
| Wearables for fall detection | 30 (13.5%) | 13 (12.6%) | 17 (14.3%) | 0.718 |
| Tremor suppression spoon | 17 (7.6%) | 5 (4.9%) | 12 (10.1%) | 0.144 |
| Tremor suppression gloves | 16 (7.2%) | 4 (3.9%) | 12 (10.1%) | 0.075 |
| Bed rails | 41 (18.5%) | 19 (18.4%) | 22 (18.5%) | 0.994 |
| Home rails | 71 (31.9%) | 29 (28.2%) | 42 (35.3%) | 0.255 |
| Bed support rails | 62 (27.9%) | 23 (22.3%) | 39 (32.8%) | 0.084 |
| Anti-slip mat | 77 (34.7%) | 39 (37.9%) | 38 (31.9%) | 0.354 |
| Cushioned fall mat | 37 (16.7%) | 14 (13.6%) | 23 (19.3%) | 0.253 |
| Horizontal lines on the floor as visual cues | 15 (6.7%) | 9 (8.7%) | 6 (5.0%) | 0.274 |
| Button-up device | 10 (4.5%) | 5 (4.9%) | 5 (4.2%) | 0.815 |
| Shower chair | 85 (38.3%) | 33 (32.0%) | 52 (43.7%) | 0.075 |
| Commode | 49 (22.1%) | 19 (18.4%) | 30 (25.2%) | 0.226 |
| Anti-choking cup | 26 (11.7%) | 7 (6.8%) | 19 (16.0%) | 0.034* |
| Fall alarm | 61 (27.5%) | 29 (28.2%) | 32 (26.9%) | 0.833 |
| Electric home ladder | 11 (4.9%) | 4 (3.9%) | 7 (5.9%) | 0.494 |
| Suction device | 10 (4.5%) | 3 (2.9%) | 7 (5.9%) | 0.287 |

All statistics were performed using the Chi-squared test for categorical units and an unpaired t-test for continuous units. A p-value less than 0.05 was considered statistically significant.

transition from HY stage 2–3 seems to be milestone in relation to gait-dependent activities and has also been found to be an index of disease progression [33].

There are several possibilities to explain our observation that patients expressed concerns about future symptoms. Firstly, they may have subtle symptoms which are not markedly manifest during an examination. It is possible that patients may actually feel these subtle symptoms developing and some patients may be even more sensitive than others to the presence of these small changes. This is similar to the finding that PD patients can have measurable subclinical tremor before visible tremor is apparent [34]. Subjective assessment of the patient's experience may therefore provide an additional advantage in helping quantify subclinical symptoms. Secondly, patients may be more concerned about future symptoms based on what they observe in other patients, for example at the clinic or of they participate in patient support groups. They may also have seen information about PD on the internet or social media, however as has been shown from a study in Korea, this material can be unreliable and even misleading [35]. Another possibility is belief in the common myth that their disease will eventually take control of them and they will become wheelchair- or bed-bound. For patients who express their concerns about future symptoms it may be of value for the physicians to explore with them why they feel this way and target management accordingly.

It is interesting to observe that none of patients at HY stage 1 reported their greatest concerns about motor symptoms, their greatest concerns related to constipation, lack of interest, and urinary problems. However, patients at HY stage 1 did share some concerns (but not greatest concerns) on a range of motor symptoms, including problems with walking and balance, difficulty speaking, and shaking but with lesser severity. It is possible that these HY stage 1 patients, who were relatively new to the diagnosis, were focusing on their most current troublesome NMS, rather than the less troublesome motor symptoms, which might have responded well to current dopaminergic medications. As far as we are aware, there is no previous information on patients' concerns in different HY stages. The closest study that we could identify reported patient's perspectives in early PD patients from the UK with up to 6 years of disease duration of which motor symptoms, including tremor, slowness, and stiffness, were rated as the most troublesome symptoms but bowel problems were also included within the top 10 most bothersome symptoms in this study [1]. Different results may reflect different study group populations. These findings should be further explored in a larger group of patients to determine a range of significant concerns that may be targets for treatment in early stage patients.

Sub-analysis by age of onset of PD found that higher concern rankings were reported by young-onset PD patients compared with typical-onset PD patients for factors that would be likely to impact on their social functioning (e.g. shaking, drug-induced dyskinesia) and ability to work, such as slow movement and difficulty performing fine finger movements. In addition, drug-induced dyskinesia was not found to be the most commonly concerning symptom in the patients surveyed in this study except for a group of young-onset PD patients who identified drug-induced dyskinesia as the most commonly concerning symptom significantly more than typical-onset patients although they did not rate it within the top three most commonly concerning symptoms related to fluctuation. This finding is similar to two previous studies involving patients from Japan and North America which reported that dyskinesia was not a concern with PD treatment in either group, whereas wearing off was a common concern in the North American patients while developing hallucinations was a common concern amongst Japanese patients [4, 6]. When an impact of dyskinesia on activities of daily living (ADLs) was evaluated, the percentage of patients reporting severe impact on ADLs by dyskinesia was less than 5%. Moderate impact was less than 30% whereas the majority reported mild or even no impact from dyskinesia [36]. Nevertheless, dyskinesia is the visible sign that might lead to social embarrassment or stigmatisation and continue to be a matter of debate on its impact on

different subgroups of PD patients. Whilst the prevalence of troublesome dyskinesia is falling in a number of recent studies, the presence of dyskinesia is still a matter of concern, particularly those with troublesome or disabling dyskinesia, or at-risk PD populations (e.g. young-onset patients) [37, 38].

Knowledge about the likely concerns of different patient subgroups is critical when managing different group of patients. In busy clinical practice, it is likely that physicians do not have sufficient time to administer a full rating scales or questionnaires. Therefore, a focused interview based on physician's understanding of the concerns of patient subgroups might have benefits for individual patients. For example, neurologists should ask patients about the presence of NMS, e.g. constipation, even when their examination shows that patient's motor symptoms are well under control. On the other hand, if a patient expresses concerns about gait and balance even though their examinations demonstrate minimal findings, this should be explored further since gait/balance symptoms are an indicator of disease progression and are strongly associated with disabilities.

Treatment-related adverse events are important as they can have an impact on PD symptoms and ultimately on a patient's quality of life, physical functioning, or disability. For example, neuropsychiatric adverse events can affect sleep, nausea and vomiting can affect absorption of medications, while orthostatic hypotension can affect balance and contribute to falls. In terms of adverse events in our study, GI issues were the most significant worry amongst all patients. GI problems are known to be common in PD patients, can affect the whole GI tract, and are likely to get worse as the disease progress, so it is important that physicians ask about such symptoms.

Results regarding the need for assistive devices showed relatively low scores for most devices and minimal differences between the subgroups analysed. None of the assistive device items included in our questionnaire was rated as a required item by more than half of the survey participants. It is possible that patients' knowledge of particular assistive devices was limited or lacking, especially regarding specialised devices such as tremor suppression spoons or gloves. This highlights the need for the timely referral of patients to appropriate occupational and physical therapists to discuss the available options. As patients rated this section based on their needs for assistive devices, the interpretation of their responses as 'needed' could have different meanings. For example, patients may indicate their needs for assistive devices based on either current symptoms or their worries for future symptoms. Likewise, patients may not express their needs for these devices as they may perceive these devices as a symbol of disability. Current evidence shows that on average only 9% of PD patients are referred to these specialities or to therapists who have the training and skills needed to undertake home safety assessments [39].

In our study, factors that predicted when patients were likely to have caregiver were difficulties with chewing and swallowing, and difficulties getting out of bed. In clinical practice, if these concerns become apparent in a patient who does not have a caregiver present, then this should signal to the healthcare team that further discussion is needed as part of their ongoing care plan to determine how they could best be supported. The presence of these kinds of functional difficulties should be explored and regularly reviewed by neurologists to assess the ability of the patient to manage independently as well as their impact on other related ADLs. It is possible that the presence of these types of concerns are early signs towards the development of complications, e.g. aspiration pneumonia in the case of swallowing difficulties. The support of caregivers for PD patients is important and valuable, and may delay or prevent such complications. There are data to show that home-based occupational therapy interventions can be beneficial when PD patients and caregivers participate together [40].

A balanced clinical judgement is critical when managing PD patients. By focusing on the particular symptoms, patients express concerns about does not mean that we should ignore other

symptoms. It is possible that patients do not report their symptoms because of their own lack of knowledge or misunderstanding. Indeed, a recent study identified barriers in communications about OFF periods between healthcare providers and PD patients who often perceived that their OFF periods were part of the disease, so could not be improved by physician's intervention [41].

In light of the growing interest in the use of patient-centred digital outcome (PCDO) measures in PD, it is likely that in the future there will be greater use of mobile technologies to capture data on patient perceptions [42, 43]. Recently, a roadmap for implementation of PDCOs measure has been proposed to facilitate the adoption of such mobile health technologies in PD to help inform clinical management [43].

By using this 50-item survey, this study provides an assessment of various aspects of PD patients' concerns, not limited to motor symptoms and NMS, but also including aspects such as treatment-related AEs, care for the advanced stage, and the need for assistive devices. It explores subjective aspects of the disease experience (e.g. level of happiness) as well as objective aspects (e.g. gait, tremor). Importantly, it reports data for an Asian PD cohort, which is currently lacking in the published literature. Previous literature on NMS suggests that cultural and ethnic background can indeed influence symptom perception [44]. Limitations of the study relate to its single-centre design and cross-sectional assessment which limits the possibility to see how these concerns may have evolved as disease progresses. Moreover, the validation of this survey is still preliminary as demonstrated by content validation, but still lacks a complete process of reliability and external validity assessment. It is also possible that certain items of the survey could be misinterpreted by patients. For example, "shaking" can be interpreted by patients as either tremor or dyskinesia. As the majority of PD patients will experience dementia, the exclusion of this patient group from the study may limit the generalisability of results. Identifying particular concerns of individual patients and then implementing an individualised approach to their PD care is critical to their wellbeing and optimal outcomes. Instruments such as the PDPC survey can help healthcare teams develop a more accurate picture of patients' perceptions of, and concerns about, their disease which will aid discussion about their ongoing management to allay any fears and help fulfil their individual desires and goals.

## Supporting information

**S1 Table. The Parkinson's Disease Patients' Concerns Survey (PDPC Survey).**
(DOCX)

**S2 Table. Comparison of the characteristics and PDPC Survey of patients with caregivers and those without.**
(DOCX)

**S3 Table. Patients' concerns on motor and non-motor symptoms: comparison between young-onset and typical-onset PD patients.**
(DOCX)

**S1 Data. Development and validation of the Parkinson's Disease Patients' Concerns Survey (PDPC Survey).**
(DOCX)

## Acknowledgments

Data checking and reviewed was provided by Dr Karen Wolstencroft, Bluewolf Communication Limited, UK.

## Author Contributions

**Conceptualization:** Roongroj Bhidayasiri, Marisa Chokpatcharavate, Onanong Phokaewvarangkul.

**Data curation:** Thanatat Boonmongkol, Yuwadee Thongchuam, Rachaneewan Plengsri, Marisa Chokpatcharavate, Onanong Phokaewvarangkul.

**Formal analysis:** Roongroj Bhidayasiri, Thanatat Boonmongkol, Yuwadee Thongchuam, Nitinan Kantachadvanich, Pattamon Panyakaew, Onanong Phokaewvarangkul.

**Funding acquisition:** Roongroj Bhidayasiri.

**Investigation:** Roongroj Bhidayasiri, Saisamorn Phumphid, Priya Jagota.

**Methodology:** Roongroj Bhidayasiri.

**Project administration:** Roongroj Bhidayasiri, Saisamorn Phumphid, Nitinan Kantachadvanich.

**Resources:** Roongroj Bhidayasiri.

**Supervision:** Roongroj Bhidayasiri.

**Validation:** Roongroj Bhidayasiri, Saisamorn Phumphid, Pattamon Panyakaew, Priya Jagota, Rachaneewan Plengsri, Onanong Phokaewvarangkul.

**Visualization:** Roongroj Bhidayasiri, Pattamon Panyakaew, Priya Jagota, Rachaneewan Plengsri.

**Writing – original draft:** Roongroj Bhidayasiri.

**Writing – review & editing:** Roongroj Bhidayasiri, Onanong Phokaewvarangkul.

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
