## [Decision Letter · Decision Letter 0]

29 Jul 2020

PONE-D-20-18129

Impact of disease stage and age at Parkinson’s onset on patients’ primary concerns: Insights for targeted management

PLOS ONE

Dear Dr. Bhidayasiri,

Thank you for submitting your manuscript to PLOS ONE. After careful consideration, we feel that it has merit but does not meet PLOS ONE’s publication criteria as it currently stands. The reviewers have raised a number of important concerns about the manuscript as it currently stands. This includes important remarks about scale validation and interpretation of results. However, we invite you to submit a revised version of the manuscript if you are able to addresses the points raised during the review process.

We look forward to receiving your revised manuscript.

Kind regards,

Mathias Toft, MD, PhD

Academic Editor

PLOS ONE

Journal Requirements:

2. Please provide additional details regarding participant consent. In the Methods section, please ensure that you have specified what type of consent you obtained (for instance, written or verbal) and whether the ethics committee approved this consent procedure. If verbal consent was obtained please state why it was not possible to obtain written consent and how verbal consent was recorded. If your study included minors, state whether you obtained consent from parents or guardians.

"No"

Reviewers' comments:

Reviewer's Responses to Questions

**Comments to the Author**

1. Is the manuscript technically sound, and do the data support the conclusions?

Reviewer #1: Partly

Reviewer #2: Partly

2. Has the statistical analysis been performed appropriately and rigorously? 

Reviewer #1: Yes

Reviewer #2: Yes

3. Have the authors made all data underlying the findings in their manuscript fully available?

Reviewer #1: Yes

Reviewer #2: Yes

4. Is the manuscript presented in an intelligible fashion and written in standard English?

Reviewer #1: Yes

Reviewer #2: Yes

5. Review Comments to the Author

Reviewer #1: Overall, this is a well written paper on patients' concerns in Parkinson's disease, an area increasingly recognized in clinical medicine to be important to delivery of good patient-centred care.

However, there are a number of points I would like the authors to address.

MAJOR COMMENTS

1. It is stated that items in the PDPCQ were rated "from 0 (not concerned) to 10 (most concerned)", but it is unclear (e.g., in Table 2) how items were ranked. The meaning and distinction between the columns "Some concerns" and "Most concerns" in Table 2 is not explained. Why was fatigue, for example, which had a mean score of 4.40 in the "Some concerns" column, ranked higher than constipation which had a mean score of 5.16? (also I note that the mean/SD figures for Fatigue and Constipation look like exact mirror images of each other and I would ask the authors to verify that this is not an error?). In Table 4, I don't understand how/why no motor symptom achieved a ranking for HYI patients - does this mean that all 6 patients with HYI rated all 12 motor symptoms as 0 ("not concerned")? This would seem highly unusual to me.

2. Similarly, regarding the statement in the Results: "Drug-induced dyskinesia was not rated by patients as a concerning symptom", does this mean that no patient gave this item a rating of at least 1/10? This seems to be at odds with the authors' statement that young-onset PD patients were concerned about drug-induced dyskinesia. This important (and perhaps controversial) point regarding dyskinesias not being an issue of patient concern was again mentioned in the Discussion and the authors should present the data in the manuscript text itself to substantiate this point.

3. I would probably tone down claims that the PDPCQ was "validated" (e.g., 1st sentence of the Discussion: "Our study using the validated PDPCQ …"), since it appears that only content validation/IOC was performed - and many other validation tests/metrics were not performed (e.g., can say "preliminary" validation was conducted. A brief inspection of the instrument reveals a number of areas that will need to be improved e.g., for future use: (i) "Shaking" without further clarification, to (presumably) denote "Tremor" (however, it is well known that patients not uncommonly say "Shaking" to describe dyskinesias); (ii) I am not sure why "Social activities" - which will commonly be influenced by many issues including non-motor features such as anxiety - should be categorized as a "motor" concern; (iii) Although it is true that "Weight loss" is more common in advanced PD, studies have shown that this is often also an early feature of PD; (iv) I am doubtful that the dichotomous choice of "Need" vs. "No need" really captures "concerns" about assistive devices, e.g., stigma associated with the use of walking aids; (v) In the "Adverse events" section, I have concerns about how patients are meant to interpret e.g., "General symptoms" or "Cardiovascular symptoms" (postural giddiness??). It is of course too late to alter the questionnaire for the purposes of publishing this paper, but some of these issues should at least be discussed as study limitations.

4. The authors should not report that a result is "statistically significant" without specifying the actual values/effect size at the same time, e.g., in the Results: "…, both of which had statistically significant odds ratios". The latter are arguably more important than statistical significance. Please rectify this elsewhere in the manuscript too.

5. For Table 3, it is unclear how many/what factors were considered for entry into the model, prior to the 2 items ("Difficulty chewing/swallowing" and "Getting out of bed") emerging as predictive factors.

MINOR COMMENTS

1. In the Results, the meaning of the statement "half of the patients … had support from caregivers". Do the authors mean that half the patients needed help from caregivers? In Table 1, I think the same thing is phrased as "Presence of caregiver" which could mean something else, e.g., that a caregiver was present during the administration of the questionnaire. The "presence of caregivers was significantly higher …" is again mentioned in the Results section, with unclear meaning.

2. In Table 1, bracket signs in the table are inverted, please correct.

3. In Table 1, the meaning of "Presenting symptoms" is unclear. Do the authors mean symptoms endorsed by the patient to be present at the time when the questionnaire was done? Or symptoms occurring at the time of initial disease presentation?

4. I am not sure that I agree with the statement: "Trying to fine-tune dopaminergic medications … to improve motor symptoms … is likely to worsen a patient's constipation". On what basis do the authors make this statement? I am not aware of good evidence indicating that levodopa worsens constipation, nor dopamine agonists except those with anticholinergic activity such as piribedil.

Reviewer #2: This is an important topic and overall the authors have done a laudable job. My main concern is with the "development and validation" of their scale which is not provided in sufficient detail and would probably be its own manuscript if done with reliable item selection and other psychometric methods. Other concerns are mainly to do with clarity. Comments by section:

ABSTRACT

- In methods it states this scale was developed and validated. It is not clear here whether this was done and previously published or if this was part of the current paper. If the goal of the current paper more details are needed.

- Unclear how a cross-sectional study could be "randomized", nor how this relatively small study of a single new scale be called "comprehensive".

INTRO

- 2nd sentence in first paragraph should have references

- If a goal of this paper is to develop and validate a new scale this should be clearly stated. If this scale was previously validated, this should be referenced.

METHODS

- What is ChulaPD?

- As I understand it, the questionnaire was developed by a movement team coming up with 50 questions to cover several areas of interest. I am not a scale development expert but this does not seem adequate - most scales go through several iterations of item development, including work with the population of interest to make sure items are representative, understandable and clinimetrically sound. It may be more fair to say you performed a cross-sectional survey and present those results than developed and validated a scale.

- Do we know how patients interpreted the term "concerns" - e.g. current problems vs. future worries vs. both, or perhaps variable between people?

- It is unclear what the domain "palliative care" means. It seems that this is used to describe advanced disease.

- It is unclear what it means that the survey was "randomly" distributed. I assume this is simply a convenience sample and that there was no sampling strategy.

RESULTS

- parentheses are backwards in Table 1

- It is unclear how "given the most concern" was determined. Was this simply scores = 10? The highest score? Using some cut-point? Whatever the method, it should be clearly described and justified.

- the "Caregiver Support" section is confusing. How was "palliative care stage" defined? The statement on "factors predictive of need for caregiver support" is misleading. As I understand it these are simply factors associated with having a caregiver.

- I find it hard to believe that HY I really had no concerns about motor symptoms. They ranked everything as 0? Perhaps this is an issue with the survey although I would believe that nonvoter concerns might outweigh motor concerns.

- Table 4 (and related text) is confusing. How was "rated 1st" determined? Most common (and if so is this simply a score above 1)? Highest mean score? Highest mean score amongst those reporting the symptoms? How is it possible that some categories have multiple items ranked 1st? Did they have exactly the same score?

- For the subgroup analyses (e.g. by age of onset) do we know if the differences between groups were statistically significant? Some of the differences mentioned seem small.

- How is early and advanced stage PD defined? How is it that over 13% of early stage PD need wheelchairs?

- In the

DISCUSSION

- This section starts by stating "the validated PDPCQ". A reference is needed. I do not think this paper is adequate to claim this is a validated scale.

- Second paragraph calls into question how patients are interpreting questions and whether most interpret them in the same way. If we do not know this basic question, or if the term in the survey is vague or could have multiple meanings it calls into question findings.

- Another big concern is the exclusion of persons with MCI and dementia in a study that claims to be "comprehensive". 75% of people with PD will develop dementia - excluding such persons limits conclusions and statements around people with advanced disease.

6. PLOS authors have the option to publish the peer review history of their article (what does this mean?). If published, this will include your full peer review and any attached files.

Reviewer #1: No

Reviewer #2: No

---

## [Author Response · Author response to Decision Letter 0]

8 Oct 2020

6 October 2020

Mathias Toft, MD., PhD.

Academic Editor

PLoS One

Dear Prof. Toft,

Re: Manuscript # PONE-D-20-18129: Impact of disease stage and age at Parkinson’s onset on patients’ primary concerns: Insights for targeted management

Thank you very much for your letter of the 30th of July 2020. We found the editor’s and reviewer’s comments very helpful and respond to them as follows:

Reviewer #1: Overall, this is a well-written paper on patients’ concerns in Parkinson’s disease, an area increasingly recognized in clinical medicine to be important to delivery of good patient-centred care. However, there are a number of points I would like the authors to address.

Major comments

1) It is stated that items in the PDPCQ were rated “from 0 (not concerned) to 10 (most concerned)”, but it is unclear (e.g. in Table 2) how items were ranked. The meaning and distinction between the columns “Some concerns” and “Most concerns” in Table 2 is not explained. Why was fatigue, for example, which had a mean score of 4.40 in the “Some concerns” column, ranked higher than constipation which had a mean score of 5.16? (also I note that the mean/SD figures for Fatigue and Constipation look like exact mirror images of each other and I would ask the authors to verify that this is not an error?). In Table 4, I don’t understand how/why no motor symptoms achieved a ranking for HY1 patients – does this mean that all 6 patients with HY1 rated all 12 motor symptoms as 0 (“not concerned”)? This would seem highly unusual to me. 

Response: We would like to apologise for the oversight of excluding this important information and would like to thank the reviewer for giving us the opportunity to revise the manuscript. Patients were asked to rate their concerns for each item on a severity scale from 0 (not concerned) to 10 (most concerned) based on their experience during the past month. Then, the severity of 0-6 was graded as ‘Some concern’ while 7-10 was considered as ‘Most concerning’. As this study focuses on the impact of disease stage and age at PD onset on patients’ primary concerns, we have revised the tables to include the results of ‘Most concerning’ as the main findings to avoid uncleared interpretation for our potential readers. 

To clarify the issue on the grading of the severity of patients’ concerns, we have revised the manuscript, which reads as follows:

(Page 7)

Therefore, our study aimed to explore patient’s greatest concern about a range of factors, such as their overall feeling of happiness, the impact of adverse events, and issues regarding palliative care and assistive devices, in addition to motor symptoms and NMS.

(Page 9)

Patients were asked to rate their concerns for each item on a severity scale from 0 (not concerned) to 10 (most concerned) based on their experience during the past month. The severity was further classified as ‘Some concern’ for the rating between 0 and 6 and ‘Most concerning’ for the rating between 7 and 10. The distinction between ‘Some concern’ and ‘Most concerning’ was determined by the consensus of the Chulalongkorn Parkinson Patients’ Support Group, represented by two authors (RP and MC), who perceived that the severity for ‘Most concerning’ should reflect the 30% end of the severity spectrum.

Table 2 has also been revised to demonstrate the findings for most concerning symptoms in various domains together with overall severity. 

With regards to table 4 where the reviewer does not understand how/why no motor symptoms achieved a ranking for HY1 patients, this is because we have only reported the most concerning symptoms in table 4 according to HY stages. As none of the motor symptoms were rated as most concerning by HY1 patients, they did not feature in this table. We have therefore revised the following statements to clarify the results of table 4 and provided further explanation in the discussion to clarify the results of concerns relating to motor symptoms rated by HY1 patients. 

(Page 18)

When patients’ greatest concerns about motor symptoms and NMS were analysed according to their HY stage, it was found that patients at HY stage 1 generally did not report great concerns about motor symptoms whilst patients at more advanced HY stages described symptoms of greatest concern relating to problems with walking and/or balance, freezing of gait, and getting out of bed (Table 4). On the other hand, patients at HY stage 1 reported greatest concern about NMS such as constipation, lack of interest or enthusiasm, or urinary problems. Overall, constipation was rated as the most concerning NMS. While patients at HY stage 1 did not report great concerns about motor symptoms, they shared some concerns on problems with walking and/or balance in 66% of patients, followed by difficulty speaking and shaking both in half of patients.

(Page 30)

It is interesting to observe that none of patients at HY stage 1 reported their greatest concerns about motor symptoms, their greatest concerns related to constipation, lack of interest, and urinary problems. However, patients at HY stage 1 did share some concerns (but not greatest concerns) on a range of motor symptoms, including problems with walking and balance, difficulty speaking, and shaking but with lesser severity. It is possible that these HY stage 1 patients, who were relatively new to the diagnosis, were focusing on their most current troublesome NMS, rather than the less troublesome motor symptoms, which might have responded well to current dopaminergic medications. As far as we are aware, there is no previous information on patients’ concerns in different HY stages. The closest study that we could identify reported patient’s perspectives in early PD patients from the UK with up to 6 years of disease duration of which motor symptoms, including tremor, slowness, and stiffness, were rated as the most troublesome symptoms but bowel problems were also included within the top 10 most bothersome symptoms in this study [1]. Different results may reflect different study group populations. These findings should be further explored in a larger group of patients to determine a range of significant concerns that may be our targets for treatment in early stage patients. 

2) Similarly, regarding the statement in the results: “Drug-induced dyskinesia was not rated by patients as a concerning symptom”, does this mean that no patient gave this item a rating of at least 1/10? This seems to be at odds with the authors’ statement that young-onset PD patients were concerned about drug-induced dyskinesia. This important (and perhaps controversial) point regarding dyskinesias not being an issue of patient was again mentioned in the Discussion and the authors should present the data in the manuscript text itself to substantiate this point. 

Response: As clarified in item 1, we reported on the most concerning symptoms with regards to symptom fluctuations. As a result, drug-induced dyskinesia was not rated amongst the top three most concerning symptoms in both young-onset or typical onset PD patients as shown in table 5. However, young-onset PD patients (33.3%) did report significantly greater symptom concern on drug induced dyskinesia than typical onset PD patients (9%). Therefore, we have included the following statement to provide the results on drug-induced dyskinesias.

(Page 22)

Even though drug-induced dyskinesia was not rated amongst the top three most concerning symptoms by either groups of patients, young-onset PD patients reported drug-induced dyskinesia as a most concerning symptom significantly more than typical-onset PD patients (33.3% vs 9%, p < 0.001). 

(Page 30-31)

In addition, drug-induced dyskinesia was not found to be a most concerning symptom in the patients surveyed in this study except for a group of young-onset PD patients who identified drug-induced dyskinesia as a most concerning symptom significantly more than typical-onset patients although they did not rate it within the top three most concerning symptoms related to fluctuation.

(Page 31)

When an impact of dyskinesia on activities of daily living (ADLs) was evaluated, the percentage of patients reporting severe impact on ADLs by dyskinesia was less than 5%. Moderate impact was less than 30% whereas the majority reported mild or even no impact from dyskinesia [36]. Nevertheless, dyskinesia is the visible sign that might lead to social embarrassment or stigmatisation and continue to be a matter of debate on its impact on different subgroups of PD patients. Whilst the prevalence of troublesome dyskinesia is falling in a number of recent studies, the presence of dyskinesia is still a matter of concern, particularly those with troublesome or disabling dyskinesia, or at-risk PD populations (e.g. young-onset patients) [37, 38].

3) I would probably tone down claims that the PDPCQ was “validated” (e.g. 1st sentence of the Discussion: “Our study using the validated PDPCQ…”), since it appears that only content validation/IOC was performed – and many other validation tests/metrics were not performed (e.g., can say “preliminary” validation was conducted). A brief inspection of the instrument reveals a number of areas that will need to be improved e.g., for future use: (i) “Shaking” without further clarification, to (presumably) denote “Tremor” (however, it is well known that patients not uncommonly say “shaking” to describe dyskinesias); (ii) I am not sure why “Social activities” – which will commonly be influenced by many issues including non-motor features such as anxiety – should be categorized as a “motor” concern; (iii) Although it is true that “Weight loss” is more common in advanced PD, studies have shown that this is often also an early feature of PD; (iv) I am doubtful that the dichotomous choice of “Need” vs “No need” really captures “concerns” about assistive devices, e.g., stigma associated with the use of walking aids; (v) In the “Adverse events” section, I have concerns about how patients are meant to interpret e.g., “General symptoms” or “Cardiovascular symptoms” (postural giddiness??). It is of course too late to alter the questionnaire for the purpose of publishing this paper, but some of these issues should at least be discussed as study limitations.

Response: We agree with the reviewer on the limitations of the current instrument. As such, we have replaced the term of “validated questionnaire” with “survey” and have addressed the limitations of the current survey as shown below.

(Page 35)

Limitations of the study relate to its single-centre design and cross-sectional assessment which limits the possibility to see how these concerns may have evolve as disease progresses. Moreover, the validation of this survey is still preliminary as demonstrated by content validation but still lacks a complete process of reliability and external validity assessment. It is also possible that certain items of the survey could be misinterpreted by patients. For example, “shaking” can be interpreted by patients as either tremor or dyskinesia. The exclusion of patients with dementia may potentially exclude conclusions or statements from patients with advanced disease.

4) The authors should not report that a result is “statistically significant” without specifying the actual values/effect size at the same time, e.g., in the Results: “…, both of which had statistically significant odd ratios”. The latter are arguably more important than statistically significance. Please rectify this elsewhere in the manuscript too.

Response: Supplementary data 3 was provided with results, including actual values and effect size. Moreover, we have revised the statistical analysis part, which is read as follows:

(Page 10)

An unpaired t-test was used to compare the differences for continuous variables where the data were fit to a normal distribution for the comparison of the above three subgroups where data was fitted to a normal distribution. In addition, an effect size for mean differences of the PDPC survey scores of PD patients with and without caregivers was calculated for Cohen’s d.

5) For Table 3, it is unclear how many/what factors were considered for entry into the model, prior to the 2 items (“Difficulty chewing/swallowing” and “Getting out of bed”) emerging as predictive factors.

Response: The following factors were included into the model, which is described as follows:

(Page 10-11)

To identify predictors of having a caregiver, binary logistic regression analysis was performed in which the presence of caregiver need was a dependent variable and twenty participant-related variables were selected to run into the logistic model as independent variables, including problems with walking and/or balance, getting out of bed, freezing of gait, constipation, daytime sleepiness, pain and/or aching, slow movement, muscle cramping, difficulty performing fine finger movements, any stiffness, marked decline in physical ability, cognitive difficulties (part of NMS), bedridden wheelchair bound, difficulty swallowing, cognitive difficulties (part of symptoms of the advance stage), age, gender, presence of postural instability, and presenting symptoms of bradykinesia. Goodness-of-fit statistics using the Hosmer–Lemeshow test helped to determine whether the model adequately described the data and indicated a poor fit if the significance value was <0.05. The logistic model was undertaken with Forward (Wald) stepwise technique in order to select the most predicable variables to determine caregiver need in PD patients. The predictors were reported as odds ratio (OR) with a p-value of <0.05 (2-tailed) considered statistically significant.

Minor comments

1) In the Results, the meaning of the statement “half of the patients…had support from caregivers”. Do the authors mean that half of the patients needed help from caregivers? In Table 1, I think the same thing is phrased as “Presence of caregiver” which could mean something else, e.g., that a caregiver was present during the administration of the questionnaire. The “presence of caregivers was significantly higher…” is again mentioned in the Results section, with unclear meaning. 

Response: To avoid unclear meaning, we have provided the definition of the “presence of caregiver” in our study as follows:

(Page 9)

The presence of caregiver in this study refers to a stable main caregiver, which is defined as any person who, without being a professional or belonging to a social support network, usually lives with the patient and, in some way, is directly implicated in the patient’s care or is directly affected by the patient’s health problem [30].

2) In Table 1, bracket signs in the table are inverted, please correct.

Response: Thanks so much for spotting these errors. We are unsure why these inverted brackets occurred in the pdf version, but not the word document when submitted. We will check when submitting the revision to ensure that there are no typos or technical errors. 

3) In Table 1, the meaning of “Presenting symptoms” is unclear. Do the authors mean symptoms endorsed by the patient to be present at the time when the questionnaire was done? Or symptoms occurring at the time of initial disease presentation?

Response: It is the presenting symptoms at the time of diagnosis and table 1 has been revised accordingly.

4) I am not sure that I agree with the statement: “Trying to fine-tune dopaminergic medications…to improve motor symptoms…is likely to worsen a patient’s constipation”. On what basis do the authors make this statement? I am not aware of good evidence indicating that levodopa worsens constipation, nor dopamine agonists except those with anticholinergic activity such as piribedil. 

Response: We have removed this statement to avoid misunderstanding.

Reviewer #2: This is an important topic and overall the authors have done a laudable job. My main concern is with the “development and validation” of their scale which is not provided in sufficient detail and would probably be its own manuscript if done with reliable item selection and other psychometric methods. Other concerns are mainly to do with clarity. Comments by section.

Abstract

1) In methods it states this scale was developed and validated. It is not clear here whether this was done and previously published or if this was part of the current paper. If the goal of the current paper more details are needed.

Response: We have changed the term Parkinson’s Disease Patients’ Concerns Questionnaire to survey on Parkinson’s Disease Patients’ Concerns (PDPC Survey) and have provided additional details on the process of construction of this survey in the methods section.

(Page 3) 

A 50-item survey on Parkinson’s Disease Patients’ Concerns (PDPC Survey) was developed by a multidisciplinary care team.

2) Unclear how a cross-sectional study could be “randomized”, nor how this relatively small study of a single new scale be called “comprehensive”. 

Response: We have revised the sentence to read as follows:

(Page 3)

This study, in an Asian cohort, provides an assessment of a wide range of PD patients’ concerns, encompassing not only motor symptoms and NMS, but also treatment-related adverse events, care in the advance stage disease, and the need for assistive devices in an Asian patient cohort.

Intro

3) 2nd sentence in the first paragraph should have references.

Response: References have been included, thank you.

4) If a goal of this paper is to develop and validate a new scale this should be clearly stated. If this scale was previously validated, this should be referenced.

Response: The aim of this study was to explore patient’s concerns about a range of factors in addition to motor symptoms and NMS. The term ‘scale’ has been revised into a survey as previously mentioned.

Methods

5) What is ChulaPD?

Response: ChulaPD is the abbreviation for the name of our institute; Chulalongkorn Centre of Excellence for Parkinson’s Disease and Related Disorders. The details have been provided as follows:

(Page 7)

A multidisciplinary care team at Chulalongkorn Centre of Excellence for Parkinson’s Disease and Related Disorders (ChulaPD, www.chulapd.org)

(Page 9)

Between March and June 2019, the PDPC Survey-Thai version was administered by one of the healthcare team to PD patients attending outpatient clinics of the ChulaPD, which is a main tertiary referral centre for PD, affiliated with Chulalongkorn University Hospital and the Thai Red Cross society.

6) As I understand it, the questionnaire was developed by a movement team coming up with 50 questions to cover several areas of interest. I am not a scale development expert but this does not seem adequate – most scales go through several iterations of item development, including work with the population of interest to make sure items are representative, understandable and clinimetrically sound. It may be more fair to say you performed a cross-sectional survey and present those results than developed and validated a scale. 

Response: We agree with the reviewer that this is a cross-sectional study with the survey to explore most concerns of Parkinson’s Disease patients. The manuscript has been revised throughout to reflect this change.

7) Do we know how patients interpreted the term “concerns” – e.g., current problems vs. future worries vs. both, or perhaps variable between people. 

Response: We have undertaken a literature review and were unable to find the scope of concerns. In our personal view, concerns could be both current problems or future worries or even something in between. We have provided our personal view as follows: 

(Page 5)

In a broad sense, concerns reflect what we believe or perceive and, in the setting of PD, could be current problems, future worries, both, or even something in between.

8) It is unclear what the domain “palliative care” means. It seems that this is used to described advanced disease.

Response: We have replaced the term “palliative care” with a more appropriate term as “care in the advanced stage”.

9) It is unclear what it means that the survey was “randomly” distributed. I assume this is simply a convenience sample and that there was no sampling strategy.

Response: The reviewer is correct and we have removed the term ‘random’ from the statement.

Results 

10) Parentheses are backwards in Table 1 

Response: This comment is similar to reviewer# 1. We are unsure why these inverted brackets occurred in the pdf version, but not the word document when submitted. We will check when submitting the revision to ensure that there are no typos or technical errors.

11) It is unclear how “given the most concern” was determined. Was this simply score = 10? The highest score? Using some cut-point? Whatever the method, it should be clearly described and justified.

Response: We would like to apologise for the oversight of excluding this important information and would like to thank the reviewer for giving us the opportunity to revise the manuscript. Patients were asked to rate their concerns for each item on a severity scale from 0 (not concerned) to 10 (most concerned) based on their experience during the past month. Then, the severity of 0-6 was graded as ‘Some concern’ while 7-10 was considered as ‘Most concerning’. As this study focuses on the impact of disease stage and age at PD onset on patients’ primary concerns, we have revised the tables to include the results of ‘Most concerning’ as the main findings to avoid uncleared interpretation for our potential readers. 

To clarify the issue on the grading of the severity of patients’ concerns, we have revised the manuscript, which reads as follows:

(Page 7)

Therefore, our study aimed to explore patient’s greatest concerns about a range of factors, such as their overall feeling of happiness, the impact of adverse events, and issues regarding palliative care and assistive devices, in addition to motor symptoms and NMS.

(Page 9)

Patients were asked to rate their concerns for each item on a severity scale from 0 (not concerned) to 10 (most concerned) based on their experience during the past month. The severity was further classified as ‘Some concern’ for the rating between 0 and 6 and ‘Most concerning’ for the rating between 7 and 10. The distinction between ‘Some concern’ and ‘Most concerning’ was determined by the consensus of the Chulalongkorn Parkinson Patients’ Support Group, represented by two authors (RP and MC), who perceived that the severity for ‘Most concerning’ should reflect the 30% end of the severity spectrum.

12) The “caregiver support” section is confusing. How was “palliative care stage” defined? The statement on “factors predictive of need for caregiver support” is misleading. As I understand it, these are simply factors associated with having a caregiver.

Response: We have replaced the term ‘palliative care’ with ‘care in the advanced stage’ and have provided the definition of advanced PD as follows:

(Page 9)

Since there is no consensus on the definition of early and advanced stages of PD, we adopted the traditional classification of early and advanced stages according to the HY scale where bilateral segmental involvement with postural and gait impairment (HY stage > 3), which also marks a clinical milestone, classifies advanced stage whereas early stage refers to those with HY stage 1-2 [31].

With regards to caregiver support, we also agree with the reviewer that those factors were associated with having a caregiver. Therefore, the following sentences are included for further clarification.

(Page 9)

The presence of caregiver in this study refers to a stable main caregiver, which is defined as any person who, without being a professional or belonging to a social support network, usually lives with the patient and, in some way, is directly implicated in the patient’s care or is directly affected by the patient’s health problem [30].

(Page 10-11)

To identify predictors of having a caregiver, binary logistic regression analysis was performed in which the presence of caregiver need was a dependent variable and twenty participant-related variables were selected to run into the logistic model as independent variables, including problems with walking and/or balance, getting out of bed, freezing of gait, constipation, daytime sleepiness, pain and/or aching, slow movement, muscle cramping, difficulty performing fine finger movements, any stiffness, marked decline in physical ability, cognitive difficulties (part of NMS), bedridden wheelchair bound, difficulty swallowing, cognitive difficulties (part of symptoms of the advance stage), age, gender, presence of postural instability, and presenting symptoms of bradykinesia. Goodness-of-fit statistics using the Hosmer–Lemeshow test helped to determine whether the model adequately described the data and indicated a poor fit if the significance value was <0.05. The logistic model was undertaken with Forward (Wald) stepwise technique in order to select the most predicable variables to determine caregiver need in PD patients. The predictors were reported as odds ratio (OR) with a p-value of <0.05 (2-tailed) considered statistically significant.

(Page 17)

The predominant concerns in the advanced stage were factors found to be predictive of having a caregiver, namely chewing and swallowing, and getting out of bed, both of which had statistically significant odds ratios (Table 3 and Supplementary data 3).

13) I find it hard to believe that HY1 really had no concerns about motor symptoms. They ranked everything as 0? Perhaps this is an issue with the survey although I would believe that nonvoter concerns might outweigh motor concerns. 

Response: We apologise for the misleading table in which we have corrected table 4 to indicate that only results of most concerning symptoms were included. None of the motor symptoms were rated as most concerning by HY1 patients. We have therefore revised the following statements to clarify the results of table 4 and provide the discussion to clarify the results of concerns on motor symptoms rated by HY1 patients. 

(Page 18)

When patients’ greatest concerns about motor symptoms and NMS were analysed according to their HY stage, it was found that patients at HY stage 1 generally did not report great concerns about motor symptoms whilst patients at more advanced HY stages described symptoms of most concerns relating to problems with walking and/or balance, freezing of gait, and getting out of bed (Table 4). On the other hand, patients at HY stage 1 reported greatest concerns about NMS such as constipation, lack of interest or enthusiasm, or urinary problems. Overall, constipation was rated as the most concerning NMS. While patients at HY stage 1 did not report great concerns about motor symptoms, they shared some concerns on problems with walking and/or balance in 66% of patients, followed by difficulty speaking and shaking both in half of patients.

(Page 30)

It is interesting to observe that none of patients at HY stage 1 reported their greatest concerns about motor symptoms, their greatest concerns related to constipation, lack of interest, and urinary problems. However, patients at HY stage 1 did share some concerns (but not greatest concerns) on a range of motor symptoms, including problems with walking and balance, difficulty speaking, and shaking but with lesser severity. It is possible that these HY stage 1 patients, who were relatively new to the diagnosis, were focusing on their most current troublesome NMS, rather than the less troublesome motor symptoms, which might have responded to current dopaminergic medications. As far as we are aware, there is no previous information on patients’ concerns in different HY stages. The closest study that we could identify reported patient’s perspectives in early PD patients from the UK with up to 6 years of disease duration of which motor symptoms, including tremor, slowness, and stiffness, were rated as the most troublesome symptoms but bowel problems were also included within the top 10 most bothersome symptoms in this study [1]. Different results may reflect different study group populations. These findings should be further explored in a larger group of patients to determine a range of significant concerns that may be our targets for treatment in early stage patients. 

14) Table 4 (and related text) is confusing. How was “rated 1st” determined? Most common (and if so is this simply a score above 1)? Highest mean score? Highest mean score amongst those reporting the symptoms? How is it possible that some categories have multiple items ranked 1st? Did they have exactly the same score?

Response: We have revised table 4. The ranking was performed based on the percentage of patients who rated that particular symptoms as most concerning. Therefore, it is possible that some symptoms have the same score.

15) For the subgroup analyses (e.g. by age of onset) do we know if the differences between groups were statistically significant? Some of the differences mentioned seem small.

Response: Effect size could not be determined in this subgroup analyses due to categorical data of these variables. However, statistical comparison was performed as shown in Supplementary data 4.

16) How is early and advanced stage PD defined? How is it that over 13% of early stage PD need wheelchairs?

Response: We provide the definition of early and advanced PD as shown below. As this study asked patients to rate their concerns on needs for assistive devices, which may be inclusive of future worries, it is possible that early PD patients rated their needs for wheelchairs based on their concerns for future symptoms. Additional statements were included for further clarification.

(Page 9)

Since there is no consensus on the definition of early and advanced stages of PD, we adopted the traditional classification of early and advanced stages according to the HY scale where bilateral segmental involvement with postural and gait impairment (HY stage > 3), which also marks a clinical milestone, classifies advanced stage whereas early stage refers to those with HY stage 1-2 [31].

(Page 32)

As patients rated this section based on their needs for assistive devices, the interpretation of their responses as ‘needed’ could have different meanings. For example, patients may indicate their needs for assistive devices based on either current symptoms or their worries for future symptoms. Likewise, patients may not express their needs for these devices as they may perceive these devices as a symbol of disability.

Discussion

17) This section starts by stating “the validated PDPCQ”. A reference is needed. I do not think this paper is adequate to claim this is a validated scale.

Response: The reviewer is correct and we have revised the sentence to read as follows:

(Page 28)

Our study using the PDPC survey found that patients’ concerns about their symptoms vary widely and depend on how they perceive their motor symptoms, NMS, fluctuations, and overall treatment experience, at different disease stages, which reflects in the results of previous studies (1, 2).

18) Second paragraph calls into question how patients are interpreting questions and whether most interpret them in the same way. If we do not know this basic question, or if the term in the survey is vague or could have multiple meanings it calls into question findings.

Response: The items of this survey was generated by the multidisciplinary team, which is also inclusive of patients and their caregivers. A positive content validation was also achieved when tested by another expert panel who were not involved with item generation. However, we also agree with the reviewer that it is possible that patients perceive their concerns in different ways as they may base their concerns for current symptoms or future symptoms or both. Nevertheless, our study aims to understand patients on their concerns, which can be subjective in some ways, but are also important for treating physicians to understand patient’s perspectives. 

19) Another big concern is the exclusion of patients with MCI or dementia in a study that claims to be “comprehensive”. 75% of people with PD will develop dementia - excluding such persons limits conclusions and statements around people with advanced disease.

Response: As we aim this survey to explore patient’s concerns based on self-administered instrument, it is important for subjects to be cognitively impact in order to provide reliable responses. However, we also recognise this as part of the limitation in which the following statements are included.

(Page 35)

Limitations of the study relate to its single-centre design and cross-sectional assessment which limits the possibility to see how these concerns may have evolve as disease progresses. Moreover, the validation of this survey is still preliminary as demonstrated by content validation but still lacks a complete process of reliability and external validity assessment. It is also possible that certain items of the survey could be misinterpreted by patients. For example, “shaking” can be interpreted by patients as either tremor or dyskinesia. The exclusion of patients with dementia may potentially exclude conclusions or statements from patients with advanced disease.

I would like to confirm that all authors have read the manuscript; the paper has not been previously published, and is not under simultaneous consideration by another journal. There is also no ghost writing by anyone not named on the author list.

There is no conflict of interest on all authors and we will take full responsibility for the data, the analyses and interpretation, and the conduct of the research. We had full access to all of the data; and that we had the right to publish any and all data, separate and apart from the attitudes of the sponsor. 

We would like to confirm that the manuscript has been reviewed by a native English speaker for style and grammatical accuracy. Thank you very much for consideration our manuscript for publication. Please let me know if there are any questions.

We are grateful to the editors and reviewers for the time and effort that they have put into helping us improve our manuscript.

Sincerely,

Roongroj Bhidayasiri

Corresponding author:

Roongroj Bhidayasiri, MD., FRCP., FRCPI.

Chulalongkorn Center of Excellence on Parkinson Disease and Related Disorders

Chulalongkorn University Hospital

1873 Rama 4 Road

Bangkok 10330

Thailand 

Tel: +662-256-4000 ext. 70701

Fax: +662-256-4630

Email address: rbh@chulapd.org

---

## [Decision Letter · Decision Letter 1]

30 Oct 2020

PONE-D-20-18129R1

Impact of disease stage and age at Parkinson’s onset on patients’ primary concerns: Insights for targeted management

PLOS ONE

Dear Dr. Bhidayasiri,

Thank you for submitting your manuscript to PLOS ONE. After careful consideration, we feel that the manuscript has substantially improved. There are only a few minor issues left, as pointed out by one of the reviewers. Therefore, we invite you to submit a revised version of the manuscript that addresses the points raised during the review process.

We look forward to receiving your revised manuscript.

Kind regards,

Mathias Toft, MD, PhD

Academic Editor

PLOS ONE

Reviewers' comments:

Reviewer's Responses to Questions

**Comments to the Author**

1. If the authors have adequately addressed your comments raised in a previous round of review and you feel that this manuscript is now acceptable for publication, you may indicate that here to bypass the “Comments to the Author” section, enter your conflict of interest statement in the “Confidential to Editor” section, and submit your "Accept" recommendation.

Reviewer #1: All comments have been addressed

Reviewer #2: (No Response)

2. Is the manuscript technically sound, and do the data support the conclusions?

Reviewer #1: Yes

Reviewer #2: Yes

3. Has the statistical analysis been performed appropriately and rigorously? 

Reviewer #1: Yes

Reviewer #2: Yes

4. Have the authors made all data underlying the findings in their manuscript fully available?

Reviewer #1: Yes

Reviewer #2: Yes

5. Is the manuscript presented in an intelligible fashion and written in standard English?

Reviewer #1: Yes

Reviewer #2: Yes

6. Review Comments to the Author

Reviewer #1: Thank you, this is a much improved version of the manuscript.

Reviewer #2: The authors have done a laudable job of improving this paper. I have only a few remaining comments:

- Now that I understand "most concerning" I think the results (including abstract) might be more accurately labeled as "most commonly concerning" (meaning item where the most people rated it as highly concerning) rather than "most concerning", which I would take to mean as highest ratings of concern.

- In Methods, it may be prudent to rename first section "Survey development and partial validation" or "Survey development and steps of validation" and/or end section by saying that "other aspects of survey development and validation were not pursued" to make it clear to readers that this is not a fully validated or clinimetrically tested instrument.

- In tables I would eliminate the "Ranked 1st" "Ranked 2nd"... The order of the items is clear. I think the "Ranked 1st" adds some confusion as it gives the impression that patients actually rank ordered their responses, which did not happen as I understand it.

- The results section 'caregiver support' should be renamed. Perhaps "concerns in advanced disease' as it really does not assess caregiver support. Also, Table 3 should be similarly renamed or at minimum state items associated with presence of a caregiver rather than "predictors" which implies a temporal and causal association.

- I would change sentence on dementia in the Limitations section to something like: "As the majority of patients with PD will experience dementia, their exclusion in this study may limit the generalizability of results."

7. PLOS authors have the option to publish the peer review history of their article (what does this mean?). If published, this will include your full peer review and any attached files.

Reviewer #1: No

Reviewer #2: **Yes: **Benzi Kluger

---

## [Author Response · Author response to Decision Letter 1]

12 Nov 2020

12 November 2020

Mathias Toft, MD., PhD.

Academic Editor

PLoS One

Dear Prof. Toft,

Re: Manuscript # PONE-D-20-18129R1: Impact of disease stage and age at Parkinson’s onset on patients’ primary concerns: Insights for targeted management

Thank you very much for your letter of the 30th of October 2020. We found the editor’s and reviewer’s comments very helpful and respond to them as follows:

Reviewer #1: Thank you, this is a much improved version of the manuscript.

Response: We would like to thank the reviewer for his/her suggestions that have improved our manuscript.

Reviewer #2: The authors have done a laudable job of improving this paper. I have only a few remaining comments:

1) Now that I understand “most concerning” I think the results (including abstract) might be more accurately labeled as “most commonly concerning” (meaning item where the most people rated it as highly concerning rather than “most concerning”, which I would like mean as highest ratings of concern. 

Response: Thanks so much for this suggestion. We have revised the manuscript to use the term ‘most commonly concerning’ instead of ‘most concerning’ in the manuscript. Moreover, we have clarified the term ‘most commonly concerning’ with the definition suggested by the reviewer. It reads as follows:

(Page 9)

Results were reported in a form of most commonly concerning symptoms, referring to what the most patients rated these symptoms as highly concerning.

2) In Methods, it may be prudent to rename the first section “Survey development and partial validation” or “Survey development and steps of validation” and/or end section by saying that “other aspects of survey development and validation were not pursued” to make it clear to readers that this is not a fully validated or clinimetrically tested instrument. 

Response: We have renamed the section as “Survey development and steps of validation” and also ended the section by stating that other aspects of survey development and validation were not performed as shown on pages 7 and 8.

3) In tables, I would eliminate the “Ranked 1st” “Ranked 2nd” …. The order of the items are clear. I think the “Ranked 1st” adds some confusion as it gives the impression that patients actually rank ordered their responses, which did not happen as I understand it.

Response: Tables 2, 4, and 5 were revised as suggested by the reviewer.

4) The results section ‘caregiver support’ should be renamed. Perhaps “concerns in advanced disease’ as it really does not assess caregiver support. Also, Table 3 should be similarly renamed or at minimum state items associated with presence of a caregiver rather than “predictors” which implies a temporal and causal association.

Response: Revision of a section name and revision of table 3 were performed as suggested by the reviewer. 

5) I would change sentence on dementia in the Limitations section to something like: “As the majority of patients with PD will experience dementia, their exclusion in this study may limit the generalizability of results. 

Response: As suggested by the reviewer, we have revised the sentence to be as follows:

(Page 34)

As the majority of PD patients will experience dementia, the exclusion of this patient group from the study may limit the generalisability of results.

I would like to confirm that all authors have read the manuscript; the paper has not been previously published, and is not under simultaneous consideration by another journal. There is also no ghost writing by anyone not named on the author list.

There is no conflict of interest on all authors and we will take full responsibility for the data, the analyses and interpretation, and the conduct of the research. We had full access to all of the data; and that we had the right to publish any and all data, separate and apart from the attitudes of the sponsor. 

We would like to confirm that the manuscript has been reviewed by a native English speaker for style and grammatical accuracy. Thank you very much for consideration our manuscript for publication. Please let me know if there are any questions.

We are grateful to the editors and reviewers for the time and effort that they have put into helping us improve our manuscript.

Sincerely,

Roongroj Bhidayasiri

Corresponding author:

Roongroj Bhidayasiri, MD., FRCP., FRCPI.

Chulalongkorn Center of Excellence on Parkinson Disease and Related Disorders

Chulalongkorn University Hospital

1873 Rama 4 Road

Bangkok 10330

Thailand 

Tel: +662-256-4000 ext. 70701

Fax: +662-256-4630

Email address: rbh@chulapd.org

---

## [Decision Letter · Decision Letter 2]

16 Nov 2020

Impact of disease stage and age at Parkinson’s onset on patients’ primary concerns: Insights for targeted management

PONE-D-20-18129R2

Dear Dr. Bhidayasiri,

We’re pleased to inform you that your manuscript has been judged scientifically suitable for publication and will be formally accepted for publication once it meets all outstanding technical requirements.

Kind regards,

Mathias Toft, MD, PhD

Academic Editor

PLOS ONE

Additional Editor Comments (optional):

Reviewers' comments:

Reviewer's Responses to Questions

**Comments to the Author**

1. If the authors have adequately addressed your comments raised in a previous round of review and you feel that this manuscript is now acceptable for publication, you may indicate that here to bypass the “Comments to the Author” section, enter your conflict of interest statement in the “Confidential to Editor” section, and submit your "Accept" recommendation.

Reviewer #2: All comments have been addressed

2. Is the manuscript technically sound, and do the data support the conclusions?

Reviewer #2: Yes

3. Has the statistical analysis been performed appropriately and rigorously? 

Reviewer #2: Yes

4. Have the authors made all data underlying the findings in their manuscript fully available?

Reviewer #2: Yes

5. Is the manuscript presented in an intelligible fashion and written in standard English?

Reviewer #2: Yes

6. Review Comments to the Author

Reviewer #2: (No Response)

7. PLOS authors have the option to publish the peer review history of their article (what does this mean?). If published, this will include your full peer review and any attached files.

Reviewer #2: **Yes: **Benzi Kluger

---

## [Editor Report · Acceptance letter]

19 Nov 2020

PONE-D-20-18129R2 

Impact of disease stage and age at Parkinson’s onseton patients’ primary concerns: Insights for targeted management 

Dear Dr. Bhidayasiri:

I'm pleased to inform you that your manuscript has been deemed suitable for publication in PLOS ONE. Congratulations! Your manuscript is now with our production department. 

Kind regards, 

on behalf of

Dr Mathias Toft 

Academic Editor

PLOS ONE